

# No sex difference in preen oil chemical composition during incubation in Kentish plovers

Marc Gilles[1], András Kosztolányi[2], Afonso D. Rocha[3,4], Innes C. Cuthill[5], Tamás Székely[6,7] and Barbara A. Caspers[1,8]

[1] Department of Behavioural Ecology, Bielefeld University, Bielefeld, Germany
[2] Department of Zoology, University of Veterinary Medicine Budapest, Budapest, Hungary
[3] Ecology in the Anthropocene, Department of Anatomy, Cell Biology and Zoology, Faculty of Sciences, University of Extremadura, Badajoz, Spain
[4] Centre for Environmental and Marine Studies (CESAM), Department of Biology, University of Aveiro, Aveiro, Portugal
[5] School of Biological Sciences, University of Bristol, Bristol, United Kingdom
[6] Milner Centre for Evolution, University of Bath, Bath, United Kingdom
[7] Debrecen Biodiversity Centre, University of Debrecen, Debrecen, Hungary
[8] JICE, Joint Institute for Individualisation in a Changing Environment, University of Münster and Bielefeld University, Bielefeld, Germany

Corresponding author
Marc Gilles, marc.gilles@live.fr

## ABSTRACT

Preen oil, the secretion from the uropygial gland of birds, may have a specific function in incubation. Consistent with this, during incubation, the chemical composition of preen oil is more likely to differ between sexes in species where only one sex incubates than in species where both sexes incubate. In this study, we tested the generality of this apparent difference, by investigating sex differences in the preen oil composition of a shorebird species, the Kentish plover (*Anarhynchus*, formerly *Charadrius*, *alexandrinus*). As both sexes incubate in this species, we predicted the absence of sex differences in preen oil composition during incubation. In the field, we sampled preen oil from nine females and 11 males during incubation, which we analysed with gas chromatography–mass spectrometry (GC–MS). Consistent with predictions, we found no sex difference in preen oil composition, neither in beta diversity (Bray-Curtis dissimilarities) nor in alpha diversity (Shannon index and number of substances). Based on these results, we cannot conclude whether preen oil has a function during incubation in Kentish plovers. Still, we discuss hypothetical roles, such as olfactory crypsis, protection against ectoparasites or olfactory intraspecific communication, which remain to be tested.

## INTRODUCTION

Most birds possess a sebaceous gland at the base of the tail, the uropygial gland (or preen gland), that produces an oily secretion (preen oil) (*Jacob & Ziswiler, 1982*). Birds spread preen oil over their plumage during preening (*Moreno-Rueda, 2017*). The chemical composition of preen oil typically consists of wax esters and other substances, such as

alcohols, aldehydes, alkanes, carboxylic acids and ketones (reviewed in *Campagna et al., 2012*; *Alves Soares, Caspers & Loos, 2024*). Preen oil is multifunctional, serving plumage maintenance, protection against ectoparasites (*e.g.*, feather degrading bacteria, eggshell bacteria, chewing lice) and waterproofing (reviewed in *Moreno-Rueda, 2017*). Preen oil is also an important source of body odour in birds (*Hagelin & Jones, 2007*) and may have odour-related functions, namely olfactory crypsis and olfactory communication (reviewed in *Grieves et al., 2022*).

The first step to investigate the potential function(s) of preen oil is to describe the variation in its chemical composition, notably seasonal changes and sex differences (*Grieves et al., 2022*). In many species, preen oil composition changes during breeding, specifically at the time of incubation and specifically in the incubating sex (*Reneerkens et al., 2007*), strongly suggesting that preen oil has a function associated to incubation. First, a function of preen oil during incubation could be protection against ectoparasites, in case incubating birds are exposed to high parasitic loads in the nest or to limit pathogenic infection of the eggs. This was shown in Eurasian hoopoes (*Upupa epops*) where only females (incubating sex) produce a dark preen oil that contains antibacterial substances during incubation, and that is smeared on the eggs to protect embryos from eggshell bacteria (*Martín-Vivaldi et al., 2009*; *Martín-Vivaldi et al., 2010*). Second, a function of preen oil during incubation could be olfactory crypsis, in case the incubating birds (and their clutch or brood) are exposed to olfactorily searching nest predators (*Reneerkens, Piersma & Sinninghe Damsté, 2005*). This is the case in shorebirds (order Charadriiformes), where preen oil composition shifts from monoesters to diesters during incubation (seasonal change in preen oil; documented in 19 sandpiper, six plover and one oystercatcher species, *Reneerkens, Piersma & Sinninghe Damsté, 2006*), solely in the incubating sex (sex-specific seasonal change in preen oil; documented in seven sandpiper species, *Reneerkens et al., 2007*). The diester preen oil secreted during incubation is less volatile than the monoester preen oil, which makes the incubating birds (or their clutch or brood) less detectable to olfactorily searching nest predators (*e.g.*, dog, *Reneerkens, Piersma & Sinninghe Damsté, 2005*). Finally, a function of preen oil during incubation could be olfactory intraspecific signalling (*e.g.*, for mate choice; "sex semiochemical hypothesis", *Grieves et al., 2022*). For example, in three passerine species (order Passeriformes) with uniparental incubation, preen oil composition differs between sexes during breeding (*Whittaker et al., 2010*; *Amo et al., 2012*; *Grieves, Bernards & MacDougall-Shackleton, 2019a*), which allows birds to discriminate the sex of conspecifics by smell (*Whittaker et al., 2011*; *Amo et al., 2012*; *Grieves, Bernards & MacDougall-Shackleton, 2019b*).

Although several shorebird species (order Charadriiformes) have been studied with regard to sex differences in preen oil (14 species), they were all studied using a fairly straightforward analytical method (*Reneerkens, Piersma & Sinninghe Damsté, 2002*; *Reneerkens et al., 2007*). This method consists in describing preen oil composition using a single categorical variable (*i.e.*, ester composition) with three categories (*i.e.*, monoesters only, mixture of monoesters and diesters, diesters only). Reducing the complexity of preen oil composition (usually hundreds of substances) to a single categorical variable is simple but effective. Indeed, this method revealed striking sex differences in preen oil during

incubation in uniparentally incubating species (diesters in the incubating sex, monoesters in the non-incubating sex), but not in biparentally incubating species (diesters in both sexes). However, subtle sex differences in biparentally incubating species may have been missed using this categorisation, and may be uncovered using more advanced methods (*e.g.*, multivariate analyses).

In this study, we sampled preen oil from female and male Kentish plovers (*Anarhynchus alexandrinus,* formerly *Charadrius alexandrinus*) during incubation, analysed their chemical composition using GC–MS, and tested for sex differences in alpha and beta diversity using multivariate statistical analyses. Given that both sexes incubate in this species (*Kosztolányi & Székely, 2002*), and assuming that this species undergoes the same sex-specific seasonal changes in preen oil composition as the other shorebird species studied (*Reneerkens, Piersma & Sinninghe Damsté, 2002*; *Reneerkens, Piersma & Sinninghe Damsté, 2006*; *Reneerkens et al., 2007*), we predicted an absence of sex differences in preen oil composition during incubation. Alternatively, sex differences in preen oil composition can be expected in case of sexual selection or other sex-dependent reason. It should be emphasized that, since we sampled preen oil only during the incubation period, our aim was not to investigate sex-specific seasonal changes and replicate studies from *Reneerkens, Piersma & Sinninghe Damsté (2002)* and *Reneerkens et al. (2007)*. Rather, we used their findings to make predictions on whether we should find sex differences during incubation in Kentish plovers.

## METHODS

### Study site and species

Fieldwork was conducted on breeding Kentish plovers at the Samouco saltpans complex (38°44′N, 8°59′W) on the south bank of the Tagus estuary, Portugal. In the study site, Kentish plovers breed on dykes of abandoned saltpans, nesting in the ground sparsely covered by pebbles, wooden planks and salt marsh vegetation, isolated or in proximity to nests of black-winged stilts (*Himantopus himantopus*) and little terns (*Sternula albifrons*) (*Rocha et al., 2016*). The population (20–76 breeding pairs) is resident and presents an extended breeding season, from early March, when males start to defend nesting sites, to the end of July. During the breeding season, mates generally re-nest with a different mate (sequential polygamy), but monogamy is also observed. Both parents incubate the eggs for a period of 25–26 days (*Kosztolányi & Székely, 2002*).

### Field methods

As part of a colour-ringing marking program, from May to June 2019, Kentish plovers were caught on their nests during incubation using walk-in funnel traps (T. Székely, A. Kosztolányi & C. Küpper, 2008, unpublished data). The birds were sexed using plumage characteristics, measured and ringed (T. Székely, A. Kosztolányi & C. Küpper, 2008, unpublished data). We collected preen oil from 20 birds (nine females and 11 males) by gently massaging the preen gland papilla with a 100 μl microcapillary and snapping the end of the microcapillary (containing the extracted preen oil) in a 2 ml glass vial with Teflon seal (Rotilabo®) while wearing nitrile gloves. For some breeding pairs, both partners of a

breeding pair could be sampled ($N = 8$ samples from four pairs), but for most pairs only a single bird was sampled ($N = 12$ samples). Samples were stored at $-20\,°C$ during seven months, before being transferred at $-80\,°C$ for seven months until analysis. The laying date of each nest was estimated by egg flotation (T. Székely, A. Kosztolányi & C. Küpper, 2008, unpublished data). Bird capture and sampling were carried out in accordance with the Portuguese Institute of Nature Conservation and Forestry (ICNF) guidelines (license N°1/2019) and no additional institutional animal care approval was required. To ensure the well-being of the birds, we took all necessary measures to minimize any stress caused by capture and handling. After capture, birds were placed inside a dark cotton bag before being ringed, measured and sampled. This procedure took less than ten minutes per bird. Birds were released immediately after sampling, they showed no sign of discomfort or stress (*e.g.,* increased respiratory rate, open mouth breathing, or closed eyes) and returned to incubate at their nest a few minutes after release.

## Chemical analysis

All samples were first defrosted and then extracted by adding 500 µl of dichloromethane as a solvent to the vials containing the microcapillary and the preen oil. After briefly vortexing each sample, we transferred 100 µl of the solution (preen oil and dichloromethane) into a glass vial (2 ml, Rotilabo®) containing a 100 µl glass inset, using a blunt point glass syringe (which was washed with dichloromethane between each sample). For chemical analyses, we performed GC-MS, using a gas-chromatograph (GC-2030, Shimadzu, Kyoto, Japan) equipped with a VF-5ms capillary column (30 m × 0.25 mm ID, DF 0.25, 10 m guard column, Varian Inc., Lake Forest, CA, USA) and helium (at a 1 ml/min flow rate) as a carrier gas, coupled to a mass spectrometer (GCMS-QP2020NX, Shimadzu) in split (1/10) mode. The settings for the gas chromatography were as follows; injection temperature: $310\,°C$, starting temperature: $150\,°C$, followed by an increase in temperature of $20\,°C$ per min until reaching $280\,°C$, followed an increase of $5\,°C$ per minute until reaching the end temperature of $310\,°C$, which was kept for 20 min. For the mass spectrometry, ion source temperature was set at $230\,°C$ and interface temperature at $310\,°C$. Seven GC blank samples (containing dichloromethane only) were analysed among the preen oil samples.

## Chromatographic data processing

GC-MS produces chromatograms, where each peak is a substance (defined by its retention time) and the area of the peak represents the abundance of the substance (see Fig. 1 for a typical chromatogram of Kentish plovers preen oil). We assumed that each peak represents a single substance, but we acknowledge that a single peak can represent multiple substances that coelute (*i.e.,* with the exact same retention time). We extracted peak retention times and areas from chromatograms using LabSolutions GCMS solution v4.52 (Shimadzu). Because the retention time of a substance can vary subtly between samples, we aligned the chromatograms using the *GCalignR* package (*Ottensmann et al., 2018*). We used the 20 preen oil samples and the seven GC blank samples for the alignment. Substances detected in GC blank samples, as well as substances detected in single samples, were removed to control for potential contamination. We verified the quality of the alignment with a

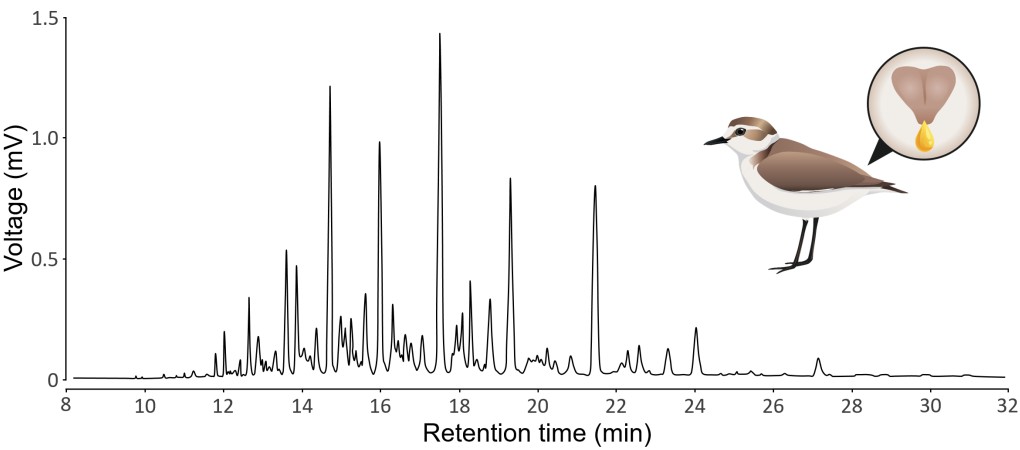

**Figure 1  Representative chromatogram of the preen oil of Kentish plovers.** The illustration depicts a female Kentish plovers with a zoom on its uropygial gland secreting preen oil.

shadeplot (Fig. S1) in PRIMER v7.0.20 (*Clarke & Gorley, 2015*). Because the amount of preen oil collected was not standardized, we used relative abundances (*i.e.,* peak area divided by total chromatogram area) for the analysis. As we have no prior knowledge about the substances potentially involved in sex differences, we log-transformed (log(X+1)) the relative abundances, thereby increasing the weight of low-abundance substances in the analysis (*Clarke et al., 2014*). We calculated the chemical diversity (Shannon index) and richness (number of substances) of each sample, as measures of alpha diversity, using the *vegan* package (*Oksanen et al., 2019*). If the detectability of substances was positively correlated to their retention time, there could be a methodological issue (*e.g.,* more volatile substances evaporating or breaking more than less volatile substances). We verified that this was not the case, as the effect of the retention time on the detectability of substances was not positive linear (polynomial beta regression: $\beta = 0.54$, $P = 0.48$), but negative quadratic (polynomial beta regression: $\beta = -5.57$, $P < 0.001$). This shows that both more volatile and less volatile substances were less detected than substances with intermediate retention time (Fig. S2). All the chromatographic data processing was conducted in R v4.2.2 (*R Core Team, 2022*) and is detailed in an R Markdown document (*Baumer & Udwin, 2015*) in the Supplemental Information.

An accurate identification of the substances would have required sophisticated analytical methods, including calculating retention indices, comparing substances with commercially available standards and using two columns of different polarity (*e.g., Alves Soares, Caspers & Loos, 2024*). For structural identification of esters, other methods could be conducted, such as combined GC and GC-MS using synthesized standards (*Sinninghe Damsté et al., 2000*; *Rijpstra et al., 2007*). As we were interested in quantitative, rather than qualitative, chemical differences, we did not need to identify the substances and used retention times instead. We putatively identified the chemical substances by comparing their mass spectrometry (MS) to that of the NIST library (NIST/EPA/NIH Mass Spectral Library 2017) and recording the substance name with the highest match, but this method is not accurate

enough to identify substances with certainty. For this reason, we do not provide the list of putative (and likely erroneous) substance names. However, we recorded the class of the substances, in case the class of the putatively identified substances was the same across all samples (see Table S1 for the list of substances, including retention times, mean relative abundances and classes). The raw chromatographic data are available at the repository PUB–Publications at Bielefeld University (https://pub.uni-bielefeld.de/record/2987587, DOI: https://doi.org/10.4119/unibi/2987587).

## Statistical analysis

We tested for sex differences in preen oil composition using 20 samples (nine females, 11 males). First, to test for sex differences in the overall chemical composition (*i.e.,* beta diversity), we performed a permutational multivariate analysis of variance (PERMANOVA) on Bray-Curtis dissimilarities using the *adonis2* function from the *vegan* package (*Oksanen et al., 2019*). Bray-Curtis dissimilarity is pertinent for the analysis of abundance data, notably because it ignores joint absences (*Clarke et al., 2014*). PERMANOVA was run with 9,999 permutations and sequential effects (type I sums of squares). As fixed effects, we included *sex*, but also *number of days after laying* to test for a potential seasonal effect as preen oil composition can change over short periods (less than a week, *Grieves et al., 2022*), and the interaction between *sex* and *number of days after laying*, as seasonal changes may differ between sexes (*Grieves et al., 2022*). Some of our samples were collected from breeding partners ($N = 8$ "paired" samples from four breeding pairs), and preen oil can be more similar within than between breeding pairs (*Gilles et al., 2024*). To deal with this possible pseudoreplication, using blocking permutations within breeding pairs is not appropriate, because in 12 cases there is only one data point from a pair (*i.e.,* only one possible choice within a pair); therefore we applied an alternative approach. First, we randomly excluded four of the "paired" samples so that we included only independent samples ($N = 16$ samples, only one sample from a breeding pair). The four excluded samples always included two females and two males so that the ratio between the sexes was not distorted (seven females and nine males). Second, we ran iterated PERMANOVAs (1,000 iterations) on the 16 samples randomly selected before each run. We report the median (and interquartile range) of the SS, $R^2$ and $F$ values from the iterated PERMANOVA runs. $P$ values were calculated as $P = \frac{\sum_{i=1}^{N_{iterations}}(1+\sum_{j=1}^{N_{perm}}I(F_j \geq F_{obs_i}))}{N_{iterations} \times (1+N_{perm})}$ where $N_{iterations}$ is the number of iterations of PERMANOVAs, $N_{perm}$ is the number of permutations per PERMANOVA, $F_j$ is the $F$-statistic for the $j^{th}$ permutation, $F_{obs_i}$ is the observed $F$-statistic in the $i^{th}$ iteration, and I(condition) is an indicator function that equals 1 if the condition is true and 0 otherwise. We also tested for a sex difference in dispersion (or variance) using the *betadisper* function from the *vegan* package (*Oksanen et al., 2019*). We used non-metric multidimensional scaling (NMDS) plots for visualization of differences in Bray-Curtis dissimilarity (beta diversity).

Second, to test for sex differences in chemical diversity (Shannon index) and richness (number of substances) (*i.e.,* alpha diversity), we performed linear models (LMM) using the *lmer* function in the *lme4* package (*Bates, 2010*). For both models, *sex* and *number of days after laying* were included as fixed effects, and *pair ID* as random effect. Using *pair ID* as a

random effect, we controlled for the potential increased similarity within breeding pairs, and thus we could include all 20 samples (nine females and eleven males). We assessed the significance of fixed effects ($\alpha = 0.05$) by checking whether their 95% confidence interval (95% CI) contained zero, using the *broom.mixed* package (*Bolker et al., 2022*). Assumptions of normality and homoscedasticity of the residuals were verified using the *performance* package (*Lüdecke et al., 2021*). All plots were created with *ggplot2* (*Wickham, 2016*), all analyses were performed in R v4.2.2 (*R Core Team, 2022*). Data and code are available in the Supplemental Information and at the repository PUB–Publications at Bielefeld University (https://pub.uni-bielefeld.de/record/2987587, DOI: https://doi.org/10.4119/unibi/2987587).

## RESULTS

We detected a total of 95 chemical substances in the preen oil of Kentish plovers, with on average 63 substances (SD = 9) per sample (on average 62 substances in females and 63 substances in males). These numbers should be treated as minima, as they are based on the assumption that one peak represents one substance, but it is possible that one peak represents multiple substances (in case of coeluting substances). Most putative substances appeared to be monoesters, while no diester was detected (Table S1). About one third of the substances (32%, $N = 35$ substances) were detected in all 20 samples, and no substance was sex-specific (*i.e.,* detected in females only or males only).

We found no sex difference in preen oil composition (beta diversity) based on Bray-Curtis dissimilarities (PERMANOVA: $P = 0.35$, $R^2 = 0.11$). The absence of a sex difference can be seen on the NMDS plot (Fig. 2A) where the 95% confidence intervals for each sex overlap entirely. The preen oil composition of females and males also did not differ in dispersion ($P = 0.39$). In addition, no sex difference was detected in alpha diversity, neither in chemical diversity (LMM: $\beta$ [95% CI] = 0.09 [$-0.15$; 0.36], Fig. 2B) nor richness (LMM: $\beta$ [95% CI] = 5.88 [$-12.7$; 26.4], Fig. 2C). Preen oil composition did not change over the course of incubation (from 1 day until 33 days after laying), neither in Bray-Curtis dissimilarities (PERMANOVA: $P = 0.48$, $R^2 = 0.11$), diversity (LMM: $\beta$ [95% CI] = 0.00 [$-0.01$; 0.01]), nor richness (LMM: $\beta$ [95% CI] = 0.02 [$-0.81$; 0.86]). Finally, we detected no effect of the interaction between sex and the number of days after laying, neither in Bray-Curtis dissimilarities (PERMANOVA: $P = 0.34$, $R^2 = 0.06$), diversity (LMM: $\beta$ [95% CI] = 0.00 [$-0.02$; 0.01]), nor richness (LMM: $\beta$ [95% CI] = $-0.32$ [$-1.73$; 0.93]). Detailed results are available in the supplemental information (Tables S2–S3).

## DISCUSSION

As predicted, we found no sex difference in the preen oil of Kentish plovers during incubation, neither in beta diversity nor in alpha diversity. This is consistent with previous findings that, in shorebirds with biparental incubation, both sexes secrete a similar preen oil during incubation (*Reneerkens et al., 2007*; *Grieves et al., 2022*). Using more advanced statistics than the classical studies on the chemistry of the preen oil of shorebirds (*Reneerkens, Piersma & Sinninghe Damsté, 2002*; *Reneerkens, Piersma & Sinninghe Damsté, 2006*; *Reneerkens et al., 2007*), we did not uncover subtle sex differences.

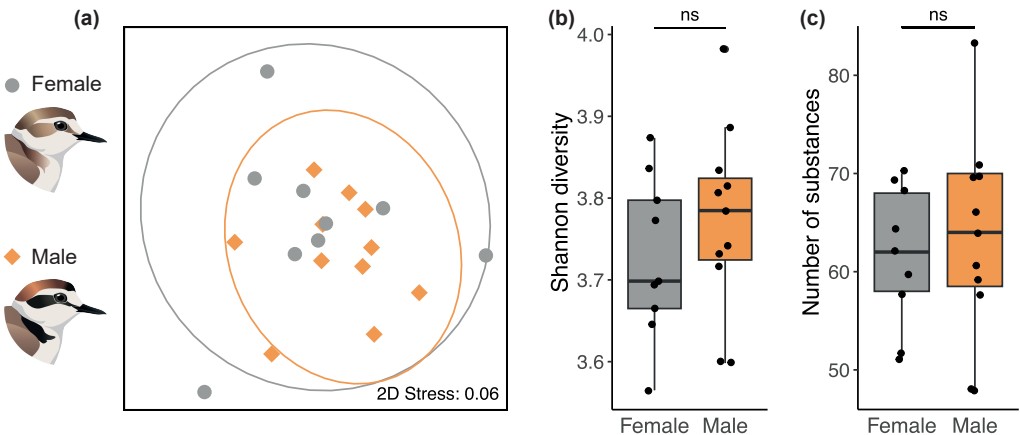

**Figure 2** **No sex difference in the preen oil composition of Kentish plovers.** (A) Non-metric multidimensional scaling (NMDS) plot representing Bray–Curtis dissimilarity in chemical composition. 2D Stress measures the "goodness of fit" of the NMDS ordination, with a value < 0.1 indicating a good fit. The ellipses for each sex (95% confidence intervals assuming a multivariate t-distribution) overlap entirely, highlighting the absence of a sex difference in beta diversity. Besides, no sex difference was detected in alpha diversity, namely (B) chemical diversity (Shannon index) and (C) chemical richness (number of substances) of preen oil.

Our finding that both sexes secrete a similar preen oil during incubation may indicate a specific function of preen oil in incubation, but only if preen oil composition changes specifically during this period, as in other shorebird species (*Reneerkens, Piersma & Sinninghe Damsté, 2002*; *Reneerkens, Piersma & Sinninghe Damsté, 2006*; *Reneerkens et al., 2007*). Because we sampled preen oil only during the incubation period, we could not test for such seasonal changes. We assumed that Kentish plover preen oil would follow the general pattern identified by *Reneerkens, Piersma & Sinninghe Damsté (2002)* in other shorebirds, that is a switch from monoesters to diesters at the onset of incubation followed by a switch back to monoesters after incubation. It seems however that Kentish plovers do not follow this general pattern. Indeed, our putative identification of the class of substances revealed that the preen oil of incubating Kentish plovers contained predominantly monoesters, and no diesters (Table S1). Although surprising, this finding is consistent with a preliminary study (*Reneerkens, 2007*), which found only monoesters in the preen oil of incubating Kentish plovers (as well as in incubating Northern Lapwings *Vanellus vanellus* and Eurasian dotterels *Anarhynchus morinellus*), and thus no seasonal change from monoesters to diesters. Together, these results suggest that, in some shorebird species including Kentish plovers, preen oil composition may not switch to a diester mixture during incubation, and challenge the idea that preen oil has a role in incubation in these species. However, even if the preen oil of Kentish plovers does not contain any diester during incubation, it may still undergo seasonal changes, although not as dramatic as a complete shift to diesters. For example, preen oil may consist of a mixture of monoesters year-round, but the monoesters produced during incubation may be less volatile than those secreted the rest of the year. However, if the seasonal changes are only subtle, they may

not affect volatility sufficiently to play a role in olfactory crypsis. In any case, we call for caution with these preliminary results, because the analytical methods used by *Reneerkens (2007)* (*i.e.,* judging peak patterns from chromatograms) and our study (*i.e.,* comparing mass spectrometry with the NIST library) are simplistic and prone to inaccuracies. This warrants a more accurate identification of the substances in the preen oil of Kentish plovers (*e.g., Rijpstra et al., 2007*; *Alves Soares, Caspers & Loos, 2024*), as well as an estimation of volatility, using samples collected throughout the year.

From our descriptive results, we cannot conclude whether preen oil has a function in incubation in Kentish plovers. Still, we can speculate on possible incubation-related functions. Preen oil may have a role in olfactory crypsis at the nest, although there are hints that the preen oil of Kentish plovers does not follow the pattern observed in other shorebirds studied (*Reneerkens, 2007*). Kentish plovers nest on the ground and are vulnerable to olfactorily searching nest predators, such as dogs, foxes, snakes and lizards (*Fraga & Amat, 1996*; *Kosztolányi et al., 2009*). When producing a low-volatility preen oil, incubating birds (and/or their clutch or brood) may be less detectable to olfactorily searching nest predators, thereby increasing nest survival (*Reneerkens, Piersma & Sinninghe Damsté, 2005*; *Grieves et al., 2022*). To further investigate this possibility, we should sample preen oil from Kentish plovers across several breeding stages (not only during incubation) and measure its volatility. Unfortunately, there is, to our knowledge, no consensual way to measure volatility, although several methods have been proposed (*e.g., Gilles et al., 2024*). Future research should thus develop a standardised method to measure volatility (or detectability) from chromatograms or from biological samples. Alternatively, one can assess differences in volatility by conducting detection trials with predators or conspecifics (*e.g.,* trained dog, *Reneerkens, Piersma & Sinninghe Damsté, 2005*). Preen oil may also protect incubating birds from feather degrading bacteria (*e.g.,* red knots *Calidris canutus, Reneerkens et al., 2008*; Eurasian hoopoes, *Ruiz-Rodríguez et al., 2009*) and their clutch from eggshell bacteria (*e.g.,* Eurasian hoopoes, *Martín-Vivaldi et al., 2010*). To test this, we should assess the antimicrobial properties of the preen oil of Kentish plovers (*e.g., Shawkey, Pillai & Hill, 2003*). Finally, preen oil may have a role in chemical signalling for mate choice. Sex recognition based on preen oil odours, like in dark-eyed juncos (*Junco hyemalis*) (*Whittaker et al., 2010*; *Whittaker et al., 2011*) and song sparrows (*Melospiza melodia*) (*Grieves, Bernards & MacDougall-Shackleton, 2019a*; *Grieves, Bernards & MacDougall-Shackleton, 2019b*), is not likely in Kentish plovers because of the absence of sex differences during incubation. To confirm this, we should also test for sex differences in preen oil before incubation, when mate choice actually occurs. Also, preen oil may have signalling roles other than sex recognition. Birds may display their genetic compatibility (*e.g.,* major histocompatibility complex) in their preen oil odours, like in black-legged kittiwakes (*Rissa tridactyla*) (*Leclaire et al., 2014*) and song sparrows (*Grieves et al., 2019c*), and they may use preen oil to assess relatedness of a potential mate (*Krause et al., 2012*; *Caspers, Gagliardo & Krause, 2015a*). It should be emphasized that preen oil could have a function for chemical protection and chemical signalling at the same time. Indeed, preen oil odours could signal greater protection of the offspring (*e.g.,* against predators *via* olfactory

crypsis, or against pathogens *via* antimicrobial activity) and thereby be sexually selected signals.

We acknowledge that our negative results (absence of sex differences) may be due to the limited sample size and thus limited statistical power (*i.e.,* false negative, or type II error). To evaluate whether our negative results are more likely false negatives or true negatives, we can compare the effect sizes of positive results from other studies with the confidence intervals from our study (*Nakagawa & Cuthill, 2007*). A study on the preen oil composition of blue tits found a significant sex difference in chemical richness, with females producing on average 38 substances more than males (*Caspers et al., 2022*). This effect size (38 substances) falls well outside our confidence interval ([−12.7; 26.4]), indicating that our study would have had the power to detect such an effect. Although this does not prove that our results are true negatives, it gives us confidence that they likely are. We note that we did not focus on the volatile fraction of preen oil, which would be the most relevant to study for its putative odour-related roles, like olfactory crypsis or chemical signalling. Instead, we analysed the whole preen oil composition, which includes both volatile and nonvolatile compounds. We did so because nonvolatile compounds may be precursors of volatile compounds involved in crypsis or signalling, and are thus also relevant for such studies (*Mardon, Saunders & Bonadonna, 2011*). For example, the monoesters and diesters in the preen oil of red knots *Calidris canutus* are nonvolatile but still seem to have different odours or odour levels (different detection success of monoester and diester preen oil by a dog, *Reneerkens, Piersma & Sinninghe Damsté, 2005*). Another example is the preen oil of song sparrows, where the sex differences in nonvolatile esters seem to translate into sex differences in body odour, allowing the birds to discriminate sex by smell (*Grieves, Bernards & MacDougall-Shackleton, 2019b*).

Our results are based on a single species and a single period, and thus cannot elucidate whether preen oil has a role (and which role) in incubation. However, our study provides valuable data on sex differences in preen oil. To investigate the function of preen oil in Kentish plovers, future studies should sample preen oil at different breeding stages (notably during mate choice and non-breeding) and measure its volatility. Importantly, future studies should conduct experiments, such as antimicrobial assays to test for antiparasitic protection (*e.g., Reneerkens et al., 2008*; *Martín-Vivaldi et al., 2010*), detectability trials or field experiments to test for olfactory crypsis (*e.g., Reneerkens, Piersma & Sinninghe Damsté, 2005*; *Selonen et al., 2022*), and behavioural trials to test for olfactory communication (*e.g., Caspers et al., 2015b*; *Grieves, Bernards & MacDougall-Shackleton, 2019b*).

## CONCLUSION

Sex differences in preen oil composition could not be detected during incubation in Kentish plovers, a shorebird species in which both sexes incubate. This result is consistent with previous studies, where sex differences in preen oil occurred during incubation in uniparentally incubating species more than in biparentally incubating species. The similar preen oil secreted by females and males during incubation may have a function for olfactory crypsis, as proposed for other shorebird species, but also for protection against ectoparasites

and/or olfactory communication, and may have no incubation-related function at all. To elucidate whether preen oil has a function in incubation, future studies should first test whether preen oil composition changes seasonally, specifically at the time of incubation.

## ACKNOWLEDGEMENTS

We are grateful to Fundação das Salinas do Samouco, Portugal for granting us access to their facilities. We also extend our gratitude to Bitrus Kwanye Zira, Xia Zhan, Emma James and Artur Silvério for their invaluable assistance with fieldwork. We thank Jeroen Reneerkens, Alice Poirier and an anonymous reviewer for comments that improved this manuscript.

### Funding

Marc Gilles was supported by Deutsche Forschungsgemeinschaft (421568765). The study was supported by a National Research, Development and Innovation Office, Hungary grant (NN 125642) awarded to András Kosztolányi. The funders had no role in study design, data collection and analysis, decision to publish, or preparation of the manuscript.

### Grant Disclosures

The following grant information was disclosed by the authors:
Deutsche Forschungsgemeinschaft: 421568765.
National Research, Development and Innovation Office, Hungary:  NN 125642.

### Competing Interests

The authors declare there are no competing interests.

### Author Contributions

- Marc Gilles conceived and designed the experiments, performed the experiments, analyzed the data, prepared figures and/or tables, authored or reviewed drafts of the article, and approved the final draft.
- András Kosztolányi conceived and designed the experiments, analyzed the data, authored or reviewed drafts of the article, and approved the final draft.
- Afonso D. Rocha performed the experiments, authored or reviewed drafts of the article, and approved the final draft.
- Innes C. Cuthill conceived and designed the experiments, authored or reviewed drafts of the article, funding acquisition, and approved the final draft.
- Tamás Székely conceived and designed the experiments, authored or reviewed drafts of the article, funding acquisition, and approved the final draft.
- Barbara A. Caspers conceived and designed the experiments, authored or reviewed drafts of the article, funding acquisition, and approved the final draft.

## Animal Ethics

The following information was supplied relating to ethical approvals (*i.e.,* approving body and any reference numbers):

Portuguese Institute of Nature Conservation and Forestry (ICNF) (license N○1/2019).

## Data Availability

The chemical (chromatographic) data, metadata, variables of the metadata of the preen oil samples and R code used for the analyses are available in the Supplemental Files and at the repository PUB–Publications at Bielefeld University, https://pub.uni-bielefeld.de/record/2987587, DOI: https://doi.org/10.4119/unibi/2987587.

## Supplemental Information

Supplemental information for this article can be found online at http://dx.doi.org/10.7717/peerj.17243#supplemental-information.

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
