# Peer review of "No sex difference in preen oil chemical composition during incubation in Kentish plovers"

_PeerJ, doi:10.7717/peerj.17243_

## Round 0.1 · original submission · Major Revisions

Dear Dr. Gilles and colleagues,

Thank you for submitting your manuscript to PeerJ. Your manuscript has been reviewed by three reviewers: as you can see, your manuscript has received mixed feedback and recommendations. Consequently, in light of this feedback as well as my own reading of the manuscript, I am recommending major revisions, and I would consider accepting the manuscript only if some major concerns raised by two of the reviewers are fully addressed:

1) The first concern is about the framing of the study. The main premise of the work, in fact, is that in many bird species, the incubating sex tends to change the chemical composition of their preen oil either as a protection against ectoparasites or as a way to hide the clutch odours from predators. Consequently, there is evidence that there are sex differences during the incubation period in preen oil chemical composition in bird species where only one of the sexes is responsible for the incubation. In this context, your study tests whether, in species where both sexes incubate, then you should not expect sex differences in preen oil composition. However, as both reviewers 1 and 2 point out, in order to test whether the incubating sex changes their preen oil composition, it is not enough to show that there are no sex differences in chemical composition but also that oil composition does change in both sexes during the incubation period compared to when the pair is not incubating their eggs. Therefore, I would recommend two options: a) to incorporate in the analysis a comparison in chemical composition between the breeding vs non-breeding season with the goal of showing seasonal changes in chemical composition; however, I appreciated that data collected during non-breeding season may not be available; reviewer 2 also refers to a PhD thesis in which the chemical composition of preen oil of several Kentish plovers does not differ between seasons; I would therefore suggest, as pointed out by reviewer 2 (b) a reframing of the introduction (and potentially discussion) in the light of this point;

2) Reviewer 1 points out that the sample size is particularly low, especially when sex differences are tested. I agree with them. I, therefore, recommend adding a power analysis in the methods showing that the sample size is enough to detect sex differences.

Please make sure you address also the minor comments raised by all three reviewers.

Reviewer 1 ·

Basic reporting

No comment.

Experimental design

Reneerkens showed that during incubation the composition of the uropygial secretion changed to be more cryptic for odour-guided predators, this change only affecting the incubating sex. However, the authors only measure the composition during incubation, so they cannot test whether, indeed, during incubation, the secretion changes in the incubating sex to reduce the risk of being predated. It is important to measure chemical composition at other moments of the breeding cycle. Actually, the authors are failing to test the hypothesis developed by Reneerkens and coauthors.

Validity of the findings

The sample size is perhaps very low. Statistical power probably is low and this is especially serious given that the hypothesis is the absence of sex differences. In this case, statistical error type I may be particularly concerning.

Additional comments

L. 47: It is better not to cite unpublished references except when essential (as in line 250), given that the reader cannot access the information. Once the article is accepted, it may be incorporated into the typescript if it is not published, still.

L. 92: Still, sex differences might occur as a consequence of sexual selection or other sex-dependent reasons.

L. 107: Please, indicate how they were sexes.

L. 112: How much time did it last from sampling to analyses? -20ºC could be insufficient if the storage lasted a long time.

Results: The study presents a relatively low sample size, which can compromise the conclusions (low statistical power, more importantly when the hypothesis matches with the statistical null hypothesis). One of the strengths of the study is providing the list of components of the uropygial secretion, but I did not see this finding. Please, provide the list of chemicals.

L. 333: Upupa epops in cursive.

L. 345: Two authors were lose: Reneerkens J, Piersma T, Damsté JSS. 2005.

·

Basic reporting

Review of “No sex difference in the preen oil composition of Kentish plovers during incubation” by Gilles et al.

This is a well-written manuscript that clearly shows that there are no differences in the preen oil composition in Kentish Plovers during incubation. Where previous research looked at (sex-related or seasonal) variation in preen wax composition in shorebirds by simply qualifying it into being monoesters-based, diester-based or a mixture of mono- and diesters based on viewing gas chromatograms, the authors also quantified the diversity of components.
The study is embedded in the relevant scientific background and contains the relevant references to existing literature. The figures are clear and beautiful and all data and code are accessible.

Experimental design

The experimental design is clear and would be easy to replicate.

The authors also have a clear hypothesis, stating that if preen oil has a function during incubation, preen oil composition of Kentish Plovers will not differ between male and females, given that both sexes incubate. However, if Kentish Plovers would excrete the same preen oil composition year-round, an absence of a sex-effect on preen oil composition will not indicate that preen oil has a (particular) function during incubation. In fact, the authors’ hypothesis is a simplified version of the hypothesis by Reneerkens et al. 2002, 2007) that a seasonal change in preen oil composition upon the start of incubation in the incubating sex(es) only, reflects a specific function of the preen oil secreted by the incubating sex(es) during incubation. There is no reason to believe that preen oil per sé has a function in incubation (as suggested in lines 28-29, line 57), if preen oil of the same chemical composition is secreted year-round. In other words, the authors assume (see lines 227-230 in the Discussion), but do not show, a seasonal change in preen oil composition upon the start of incubation in Kentish Plovers.

However, I showed that Kenthish Plovers, like Lapwing Vanellus vanellus and Eurasian Dotterel Charadrius morinellus do not show seasonal variation in preen wax composition (i.e. upon incubation do NOT change from a misxture of monoesters to a mixture of diesters ). I reported this in my PhD thesis on page 92, where I wrote: "The preen wax composition of thirteen Kentish plovers caught in Oregon, USA, between 30 May – 29 July 2001 all appeared to consist of monoesters only, even though eleven of the birds were caught during incubation and two after hatch."

Of course, I realize that this information may easily have been missed by the authors, as it is only available in my PhD thesis, but was never published in a peer-reviewed journal. Still, that the chemical data (raw data) did not show any diester (if I am correct), which is the kind of substance produced by most incubating shorebirds when seasonal variation in preen oil composition has been described, but predominantly monoesters (commonly produced year-round by shorebirds that do not incubate), could have been a sign that something was "atypical" and would have deserved to be mentioned.

Thus, in this study speculation about the function of preen oil in Kentish Plovers should go beyond functions during incubation only.

If of interest, a PDF of my PhD thesis is available here: https://research.rug.nl/files/279480

Validity of the findings

no comment

Additional comments

Minor comments:

L195 The authors indicate that they found 95 substances in the preen oil by Kentish Plovers, but this is based on the assumption that each peak in a gas chromatogram represents a specific substance. However, different substances can have the same retention time, so this is a minimal amount of substances. A more thorough analysis using GC-MS-MS (e.g. (Sinninghe Damsté et al. 2000, Rijpstra et al. 2007) would help to identify which substances are included the preen oil.

L195-197 Clearly, there are some (subtle) inter-individual differences in the preen oil composition in Kentish Plovers. Although this was not the focus of the authors’ study, it would be worthwhile to spend a few words on it. I wonder whether it was some specific substances that were present in all samples, and which substances were not. This could point at a simple methodological issue (e.g. the most volatile components, or those that easily break-down may not always be sampled and detected) or a functional issue (e.g. it may be related to individual condition whether a specific substance is secreted). The nice (!) figure S1, seems to suggest that most inter-individual variation exists in the most volatile components (i.e. those with a short retention time)?

L213-215 “This extends… during incubation (Reneerkens et al. 2007, Grieves et al. 2022)”. While this statement is correct in itself, it ignores that this is related to (and assuming) a seasonal change in preen oil composition. In that sense, the comment that “adding a new species to the list” (L 218) incorrectly suggests that previous studies focused on sex differences in preen oil composition during incubation, while instead they focused on finding sex differences in seasonal changes in preen oil composition.

I would encourage the authors to also briefly report on the kind of substances found in Kentish Plover preen oil (monoesters, diesters, range of carbon chain length etc) and describe how they were identified. This may be of interest for future researchers comparing preen oil composition between species.

·

Basic reporting

This manuscript investigates sex differences in preen oil chemical composition of Kentish plovers during incubation. Although the research question, and corresponding data, are rather straightforward, the authors do a good job of including their study both in the existing body of literature on the subject (e.g. they directly compare their work to previous one looking at the same question but with different statistical analyses), and in the larger framework of animal chemosignalling (e.g., by discussing all the putative roles of preen oil in these birds). I believe this manuscript can make a useful contribution to the field of animal chemosignalling and behaviour.

The manuscript is very well written and of reasonable length, I found it very enjoyable to read. Adequate referencing was made throughout. I made a few suggestions regarding language and referencing in the General Comments section below, but these are small.

The introduction does a good job of presenting what is known of bird preen oil composition and its putative functions. The authors justify their research question by adequately mentioning previous research conducted using a more simplistic methodology of volatile categorization. The hypothesis is clear and concise; the results answer it in a self-contained way; and the discussion provides a good summary of the results as well as ideas for future research on bird chemosignalling.

The figures are both clear and aesthetically pleasing. The raw data and supplementary material are self-explanatory and presented in a clear way.

Experimental design

The research presented in this manuscript is original; the research question is straightforward yet well justified by the new angle taken here, the use of different statistics. The manuscript is looking at validating previous findings on other bird species using multivariate statistics. The knowledge gap is clearly identified in the introduction, and well addressed in the discussion. I made small suggestions to add more details on some points of the discussion in the General Comments.

The investigation was conducted rigorously and ethically. I have made minor suggestions on the Methods section in the General Comments, to improve reproducibility and clarity of the manuscript.

The statistical approach is generally well described. I liked the idea of running iterated Permanovas on randomly selected subsets of the data and reporting the median value, to avoid the problem of pseudoreplication.

I was a little confused by the LMM results reported in Table S2. I would expect to see the estimates, std error and t-stat reported as well. I am curious to have an explanation of why the authors only report the beta and CI, mostly for my own comprehension.

Validity of the findings

The findings are valid, adding to the existing body of knowledge on bird preen oil chemical composition and its variation.

The interpretations of the data are robust, and the raw data are made accessible in a repository. The R code is clear.

The conclusions made are appropriate, and the discussion adequately opens on future directions and limitations. I provided small suggestions to broaden the discussion.

Additional comments

General Comments:

Title – I would suggest adding “chemical” to “preen oil composition”. This would make the manuscript more visible when searching for animal chemosignalling or to a more biochemistry-oriented audience.

Intro – The review by Grieves et al 2022 is cited A LOT. I agree that this is a good review, perfectly aligned with the topic of the present manuscript, however I would like to see more reference to original research instead whenever possible, which can probably be taken from the ref list cited in Grieves et al. 2022 for the most part.

L 61 – I would suggest replacing “and that” with “which”, which would be a bit more formal.

LL 65-68 – I am confused by this sentence, it is not clear what the 26 bird species mentioned, then the 7 other ones mentioned, make reference to. Do the first 26 differ between incubation periods, and the 7 others across the sexes? Please rephrase.

LL 71-72 – I would suggest adding “intraspecific”, to give more precision about the type of olfactory signalling the authors are referring to here. Consider adding one sentence to introduce further the “sex semiochemical hypothesis” before citing Grieves et al.’s review.

L 87 – I suggest replacing “method” with “categorization”.

L 102 – I suggest replacing “Within” with “During”.

L107 – Please add a reference for the walk-in funnel traps, for someone who would be interested to see how they look like.

LL 108-110 – Please add details on PPE and other measures taken to ensure there was no contamination of the odor samples.

L 124 – Please add details on the type of pipette used to transfer the solution into the vial (again thinking of the risk of contamination).

LL 126-131 – Add the width and inner diam of the column, helium flow rate, and MS parameters used. See similar publications for an example.

LL 141-142 – Please add a figure caption for the suppl figure (unless I missed it), to help the reader understand it.

L 195 – If the authors are comfortable enough with the level of compound identification achieved in their study, I would suggest adding a Suppl Table listing the 95 compounds retrieved from preen oil; these may be useful to use as reference in future studies.

LL 235-240 – I would suggest that the authors briefly state how they would assess the antimicrobial and semiochemical properties of preen oil, similar to the end of the previous paragraph discussion differences across breeding stages.

LL 245-248 – Nice addition.

L 253 – This is the only mention of volatility of the compounds recovered from preen oil. Because they are in an oil base, I expect these compounds to be mostly involatile. Yet for the purpose of investigating putative semiochemical roles of preen oil in this species, a future direction should be to look at volatile compounds too. This should be mentioned as a limitation of the current study, and could be expanded a bit as a future direction.

L 253 – The limited sample size in this study should be mentioned as a limitation.

L 261 – I suggest replacing “where” with “in which”.

L 263 – “more than”, or rather not at all?

---

## Round 0.2 · Minor Revisions

Dear Dr Gilles and colleagues,

Many thanks for revising your manuscript according to my and the reviewers' comments. Following my own reading of the revision as well as the comments from one of the original reviewers, I am happy to recommend your manuscript for publication, pending some minor revisions raised by the reviewer, which you can find in the attached document.

·

Basic reporting

I notice that the authors stress that there should be a function of preen oil during incubation. They use two final sentences in the abstract on this, but these are speculations (as clearly mentioned in the discussion) and do not follow from the results from their study. Could it not be possible that the preen oil has no special function during incubation? I would argue that based on the findings in my studies on shorebirds, that it is not so likely that there is a special function of preen oil in Kentish Plovers. If Kentish Plovers show any seasonal changes in preen oil composition, they must be rather subtle and unlikely involve large changes in volatility or anti-microbial effects.

More detailed (minor) comments are in the uploaded PDF.

Experimental design

No comments.

Validity of the findings

See my earlier comments on conclusions about the function of preen oil during incubation.

---

## Round 0.3 · accepted · Accept

Dear Dr Gilles and colleagues,

Many thanks for revising the manuscript. After reviewing your revision and your response to the reviewer, I am happy to accept the current version of the manuscript. Therefore, the manuscript is ready for publication.
Thank you again for submitting your work to PeerJ.